# Preclinical and clinical obesity: prevalence, associations to cardiometabolic risk and response to lifestyle intervention in NHANES and the EPIC-Potsdam and TULIP studies

Catarina Schiborn[1,2], Frank B. Hu[3,4,5], Norbert Stefan [2,6,7,9] &
Matthias B. Schulze [1,2,8,9] ✉

An expert commission (The Lancet Diabetes & Endocrinology Commission on Clinical Obesity) proposed novel diagnostic criteria distinguishing between preclinical and clinical obesity and suggesting treatment indications for the latter. However, the proportional assignment to preclinical and clinical obesity in adults with BMI-defined obesity, the associated disease risks, as well as the response to lifestyle interventions are not well known. Here we show that among those with BMI-based obesity, 100% are confirmed to have obesity by at least one other anthropometric measure in NHANES 2017-2018 and the prospective EPIC-Potsdam cohort. More than 80% of adults with confirmed obesity meet the criteria for clinical obesity and have 2.8-fold increased risk of incident cardiovascular disease and 7.9-fold increased risk for type 2 diabetes compared to adults without obesity and not fulfilling clinical criteria. Adults with preclinical obesity have no elevated cardiovascular disease risk, but type 2 diabetes risk is markedly increased. A 9-months lifestyle intervention (Tübingen Lifestyle Intervention Programme) decreases the proportion of clinical obesity from 71% to 57%, and that of prediabetes from 52% to 29%.

Obesity is a leading cause of disability and mortality[1]. While obesity has been associated with increased risk of various health problems, the question of whether obesity per se reflects a disease has been a matter of debate. A *Lancet Diabetes & Endocrinology Commission on the definition and diagnostic criteria of clinical obesity* (hereinafter referred to as '*Commission*') has recently proposed diagnostic criteria for preclinical and clinical obesity[2]. As a first step, diagnosing obesity requires supplementary anthropometric measures or direct body fat assessment to confirm a BMI-based classification. We refer to "confirmed obesity" throughout the manuscript to reflect this initial step. Individuals with confirmed obesity can subsequently be classified into "clinical obesity" if signs, symptoms, or diagnostic tests show abnormalities in the function of one or more tissue or organ systems due to obesity, or if substantial limitations of daily activities due to obesity exist. Individuals with confirmed obesity who do not meet criteria for clinical obesity are categorized as having "preclinical obesity". The attempt to address the contentious issue of "obesity as a disease" aims to enhance treatment guidance and reduce social stigma

[1]Department of Molecular Epidemiology, German Institute of Human Nutrition Potsdam-Rehbruecke, Nuthetal, Germany. [2]German Center for Diabetes Research (DZD), München-Neuherberg, Germany. [3]Department of Nutrition, Harvard T.H. Chan School of Public Health, Boston, MA, USA. [4]Department of Epidemiology, Harvard T.H. Chan School of Public Health, Boston, MA, USA. [5]Channing Division of Network Medicine, Department of Medicine, Brigham and Women's Hospital and Harvard Medical School, Boston, MA, USA. [6]Department of Internal Medicine IV, University Hospital Tübingen, Tübingen, Germany. [7]Institute of Diabetes Research and Metabolic Diseases (IDM) of the Helmholtz Centre Munich, Tübingen, Germany. [8]Institute of Nutritional Science, University of Potsdam, Nuthetal, Germany. [9]These authors contributed equally: Norbert Stefan, Matthias B. Schulze. ✉e-mail: mschulze@dife.de

but presents several challenges. First, in the general population high BMI strongly correlates with elevated body fatness and abdominal obesity[3,4] and recent data from US adults suggest that almost all with BMI-based obesity are confirmed to have obesity[5], which questions the requirement to confirm a BMI-based obesity classification by additional measure of body fatness. Second, individuals with high BMI often exhibit metabolic or clinical manifestations; for example, only 10% of those classified as having obesity by BMI in the National Health and Nutrition Examination Survey (NHANES) III lacked any component of the metabolic syndrome[6]. If additional obesity-related symptoms are considered to diagnose clinical obesity, few individuals with obesity would be categorized as having preclinical obesity. However, representative estimates on relative proportions of preclinical and clinical obesity are so far lacking. Also, implementing the new classification in clinical practice requires empirical data supporting its value for long-term health outcomes. Consequences of classifying obesity into preclinical and clinical obesity in terms of cancer risk have recently been investigated[7,8], but remain unknown for cardiovascular disease (CVD) and type 2 diabetes (T2D) risk. Clinical obesity may harbor subgroups of substantially different cardiometabolic risk and different requirements and goals of intervention, but evidence necessary for further staging and assignment of treatment is unavailable yet. To characterize respective responses to weight-loss and

evaluate the parameters that predict the remission of clinical obesity, data from weight loss intervention studies are required. This information is critical for evaluating the clinical and health benefits of implementing these diagnostic criteria in practice.

In this work, we provide U.S. -nationally representative estimates of the proportion of preclinical and clinical obesity among adults, analyze consequences of this classification for future CVD and T2D risk and investigate response patterns to a lifestyle intervention.

## Results

### Analyses framework and study characteristics

We investigated confirmation of a BMI-based obesity classification by other anthropometric measures and the proportion of individuals with confirmed obesity classified as clinical in NHANES 2017-2018 and the European Prospective Investigation into Cancer and Nutrition (EPIC)-Potsdam study (Fig. 1a). NHANES participants ($n = 5265$) had a mean age of 48.3 years, 48.1% were men, 62.2% non-Hispanic White, and 11.5% non-Hispanic Black. EPIC-Potsdam participants were largely women (61%) with a mean age of ~50 years and were exclusively white Europeans (Supplementary Table 1). We linked the clinical obesity status to future cardiometabolic disease risks using case-cohorts nested within EPIC-Potsdam (Fig. 1b). We furthermore investigated the potential remission from clinical obesity by lifestyle intervention in the

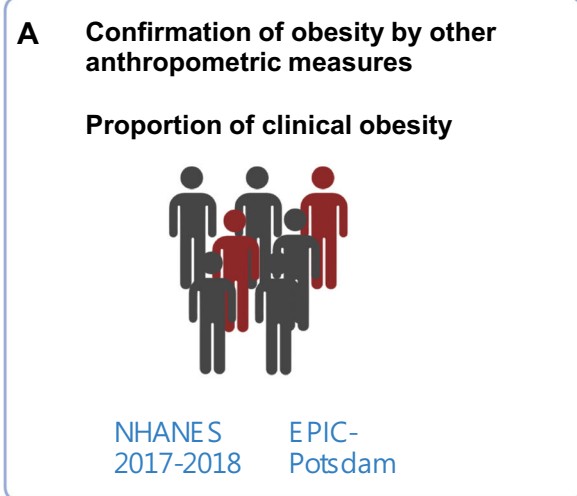

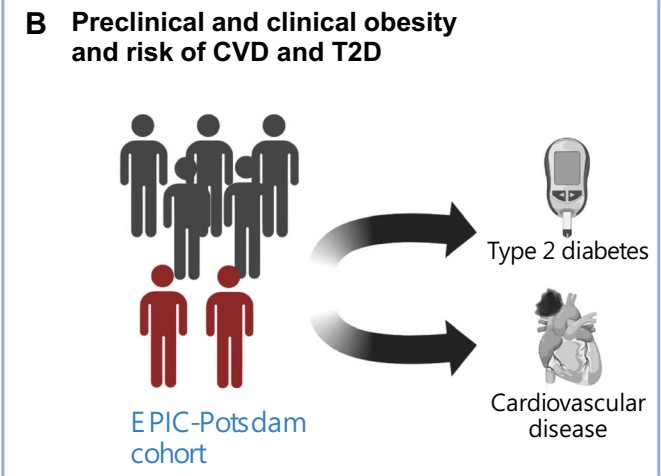

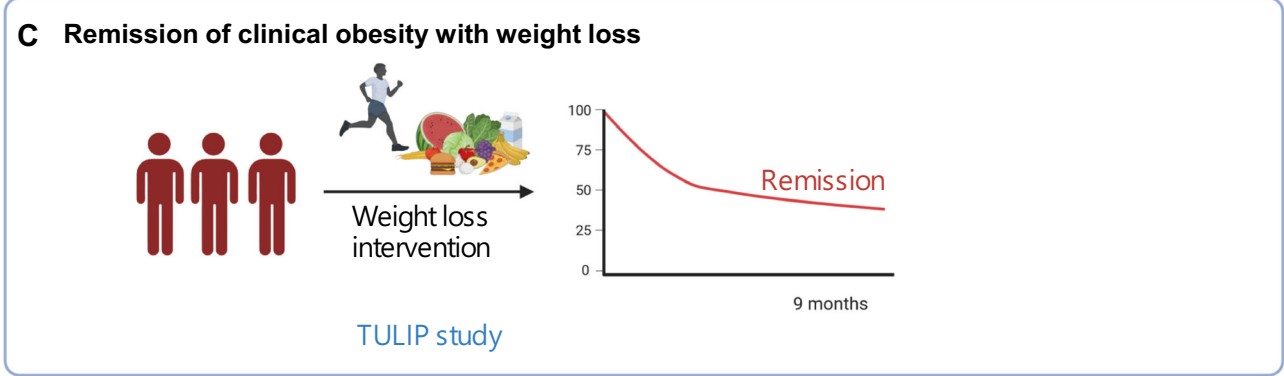

Fig. 1 | Analyses framework to characterize prevalence of clinical obesity, its association with CVD and T2D risk, and its response to a lifestyle intervention. A Proportional quantification of participants with BMI-defined obesity that fulfill at least one additional anthropometric criterion, and of participants that fulfilled the criteria for clinical obesity in the NHANES 2017-2018 survey cycle and in the EPIC-Potsdam cohort. B Association of preclinical and clinical obesity with prospective risk for CVD and T2D in the EPIC-Potsdam cohort. C Proportional quantification of

clinical obesity before and after a 9-months lifestyle intervention in the TULIP study. Abbreviations: CVD, cardiovascular diseases; EPIC, European Prospective Investigation into Cancer and Nutrition; NHANES, National Health and Nutrition Examination Survey; T2D, type 2 diabetes; TULIP, Tübingen Lifestyle Intervention Programme. Created in BioRender. Schulze, M. (2026) https://BioRender.com/c24y26m.

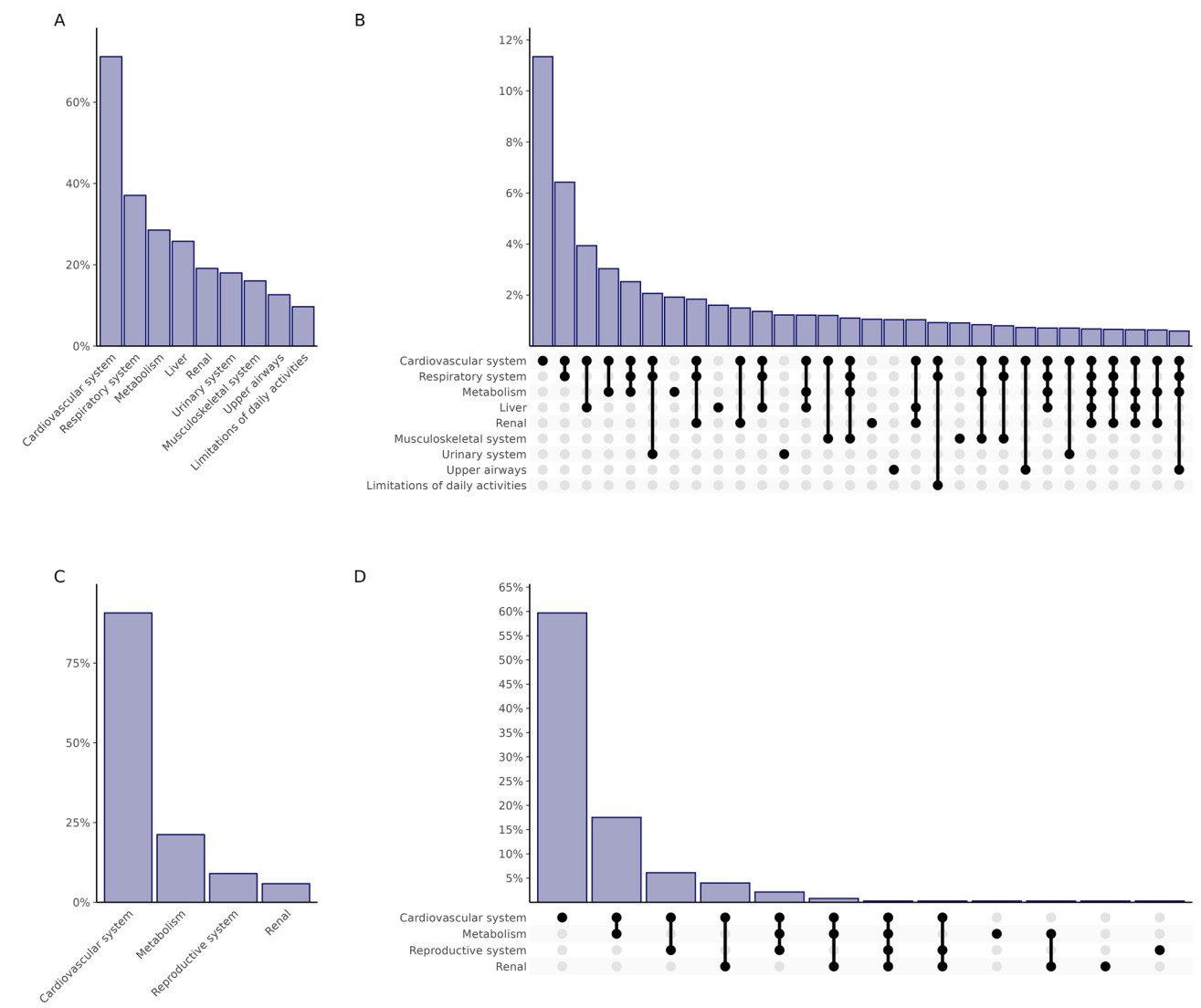

**Fig. 2 | Distribution and overlap of fulfillment of diagnostic criteria of clinical obesity in adults with confirmed obesity. A** Distribution in NHANES 2017-2018. **B** Overlap in NHANES 2017-2018. **C** Distribution in EPIC-Potsdam. **D** Overlap in EPIC-Potsdam. EPIC, European Prospective Investigation into Cancer and Nutrition; NHANES, National Health and Nutrition Examination Survey.

Tübingen Lifestyle Intervention Programme (TULIP) study over 9 months (Fig. 1c).

#### Confirmation of BMI-based obesity by further anthropometry
Among adults with obesity defined by BMI, the proportions being confirmed by individual additional body fat measures ranged between 91.6% (dual-energy X-ray absorptiometry, DXA) and 99.9% (waist-to-height ratio) in NHANES 2017-2018 and from 73.1% (waist-to-hip ratio) and 100% (waist-to-height ratio) in EPIC-Potsdam. Obesity status was confirmed by at least one additional measure in almost all adults with BMI-based obesity (Supplementary Table 2). The proportion of BMI-based obesity confirmed by DXA or waist circumference was higher in the higher BMI categories, with almost all with a BMI-based class II or III obesity being confirmed by waist circumference.

#### Proportional fulfillment of individual clinical obesity criteria
The estimated proportion of US adults (NHANES 2017-2018) with confirmed obesity fulfilling individual diagnostic criteria of clinical obesity ranged from 10.6% (95% CI, 8.7%–12.6%) for the activities of daily living (ADL) criteria to 71.1% (95% CI, 67.8%–74.4%) for the cardiovascular criteria (Fig. 2). While 18.9% of adults with confirmed

obesity fulfilled just a single diagnostic criterion, 20.1% fulfilled 2, 16.9% fulfilled 3 and even 6.9% fulfilled 6 or more criteria simultaneously (Supplementary Table 3). Overlap of adults fulfilling different diagnostic criteria was largest for the cardiovascular and respiratory criteria in NHANES (Fig. 2). In EPIC-Potsdam, 90.7% fulfilled the cardiovascular component, while 21.2% fulfilled the metabolic component. Overlap was highest for the cardiovascular and metabolic component (20.7%) (Fig. 2). The proportion of adults with confirmed obesity fulfilling individual clinical criteria of clinical obesity varied across age, BMI, gender, and race and ethnic subgroups in NHANES (Supplementary Table 4).

#### Prevalence of clinical obesity
The estimated prevalence of clinical obesity, thus of individuals with confirmed obesity who met at least one clinical criterion, in the US adult population was 36.0% (95% CI, 33.2–38.9%) and of preclinical obesity 7.6% (6.2–9.0%) (Table 1). The proportion among adults with confirmed obesity in the US who fulfilled any diagnostic criterion of clinical obesity was 82.6% (79.8–85.4%). This proportion increased with increasing BMI (class I obesity: 78.7%, class II obesity: 84.3%, class III obesity: 90.5%) and age (age 20–44 years: 69.0%, 45–64 years: 89.5%,

**Table 1 | Prevalence of confirmed obesity, preclinical obesity and clinical obesity among adults aged ≥ 20 years and proportion of clinical obesity among individuals with confirmed obesity across subgroups, NHANES 2017-2018 and EPIC-Potsdam**

| | Adults aged ≥ 20 years | | | | Adults aged ≥ 20 years with confirmed obesity | |
| --- | --- | --- | --- | --- | --- | --- |
| | No. of participants[a] | Prevalence, % (95% CI or n)[b] | | | No. of participants[a] | Proportion of clinical obesity, % (95% CI or n)[b] |
| | | Confirmed obesity | Preclinical obesity | Clinical obesity | | |
| **NHANES 2017-2018** | | | | | | |
| All adults | 5265 | 43.6 (40.5–46.8) | 7.58 (6.20–8.95) | 36.0 (33.2–38.9) | 2333 | 82.6 (79.8–85.4) |
| **Age group** | | | | | | |
| 20-44 years | 1976 | 42.3 (38.4–46.3) | 13.1 (10.8–15.5) | 29.2 (26.0–32.3) | 869 | 69.0 (64.5–73.5) |
| 45-64 years | 1897 | 46.2 (42.4–50.0) | 4.84 (2.73–6.94) | 41.3 (37.7–45.0) | 898 | 89.5 (85.2–93.9) |
| ≥ 65 years | 1392 | 42.0 (37.0–46.9) | 0.41 (0.00–0.82) | 41.5 (36.6–46.5) | 566 | 99.0 (98.1–100) |
| **Gender** | | | | | | |
| Women | 2724 | 43.3 (39.6–47.0) | 7.67 (5.73–9.61) | 35.6 (31.6–39.6) | 1260 | 82.3 (77.7–86.8) |
| Men | 2541 | 43.9 (39.1–48.8) | 7.47 (5.24–9.70) | 36.5 (33.0–39.9) | 1073 | 83.0 (79.2–86.8) |
| **Race and ethnicity** | | | | | | |
| Mexican American | 698 | 51.4 (48.0–54.9) | 14.4 (10.7–18.1) | 37.0 (33.4–40.6) | 347 | 72.0 (65.7–78.3) |
| Other Hispanic | 496 | 37.0 (32.7–41.3) | 8.11 (4.26–12.0) | 28.9 (23.2–34.6) | 199 | 78.1 (67.5–88.7) |
| Non-Hispanic White | 1807 | 42.9 (38.1–47.6) | 6.15 (4.41–7.90) | 36.7 (32.8–40.7) | 793 | 85.7 (82.2–89.1) |
| Non-Hispanic Black | 1240 | 49.1 (46.2–52.0) | 7.80 (5.49–10.1) | 41.3 (38.1–44.6) | 614 | 84.1 (79.7–88.5) |
| Non-Hispanic Asian | 759 | 34.2 (30.1–38.2) | 9.41 (7.07–11.7) | 24.8 (22.0–27.5) | 250 | 72.5 (67.8–77.1) |
| Other Race - Including Multi-Racial | 265 | 46.6 (36.0–57.3) | 10.0 (5.11–14.9) | 36.6 (26.0–47.2) | 129 | 78.5 (68.0–89.0) |
| **BMI group [c]** | | | | | | |
| Obesity class I | 1237 | NA | NA | NA | 1235 | 78.7 (75.0–82.4) |
| Obesity class II | 614 | NA | NA | NA | 614 | 84.3 (79.8–88.8) |
| Obesity class III | 483 | NA | NA | NA | 483 | 90.5 (86.4–94.6) |
| **EPIC Potsdam** | | | | | | |
| Overall | 2391 | 16.5 (394) | 1.3 (31) | 15.2 (363) | 394 | 92.1 (363) |
| **Age groups** | | | | | | |
| 20-44 years | 799 | 11.3 (90) | 1.5 (12) | 9.8 (79) | 90 | 86.7 (78) |
| ≥ 45 years | 1592 | 19.0 (304) | 1.2 (19) | 17.9 (285) | 304 | 93.8 (285) |
| **Sex** | | | | | | |
| Women | 1460 | 16.0 (233) | 1.5 (22) | 14.5 (211) | 233 | 90.6 (211) |
| Men | 931 | 17.3 (161) | 1.0 (9) | 16.3 (152) | 161 | 94.4 (152) |
| **BMI categories [c]** | | | | | | |
| Obesity class I | 315 | NA | NA | NA | 315 | 91.4 (288) |
| Obesity class II | 58 | NA | NA | NA | 58 | 93.1 (54) |
| Obesity class III | 21 | NA | NA | NA | 21 | 100 (21) |

*BMI* body mass index, *EPIC* European Prospective Investigation into Cancer and Nutrition, *NHANES* National Health and Nutrition Examination Survey.
[a]For NHANES 2017-2018: unweighted number of participants, decimals resulting from multiple imputations were rounded to the nearest integer.
[b]Weighted percent in NHANES 2017-2018 is representative of the total civilian noninstitutionalized US population aged ≥ 20 years.
[c]in non-Hispanic Asian obesity class I defined as BMI 27.5–< 32.5 kg/m², obesity class II as BMI 32.5– < 37.5 kg/m², obesity class III as BMI ≥ 37.5 kg/m² and in remaining race/ethnicity obesity class I defined as BMI 30.0–34.9 kg/m², obesity class II as BMI 35.0–39.9 kg/m², obesity class III as BMI ≥ 40.0 kg/m².

≥ 65 years: 99.0%), was comparably high in females (82.3%) and males (83.0%), and was higher in non-Hispanic Whites (85.7%) and non-Hispanic Blacks (84.1%) than other ethnic/racial groups. In EPIC-Potsdam, the proportion of confirmed obesity being clinical was 92.1%, which increased with higher BMI and age, similar to NHANES (Table 1).

Among adults without confirmed obesity, the estimated proportions fulfilling one or more diagnostic criteria for clinical obesity were 63.3% in NHANES 2017-2018 and 72.6% in EPIC-Potsdam, which differed depending on BMI (Supplementary Table 5).

**Impact of varying the metabolic component**
While the *Commission* defined the metabolic component as "cluster of hyperglycemia, high triglyceride, and low HDL-cholesterol"[2], this definition likely excludes many individuals with metabolic

consequences of obesity. We therefore repeated the analyses, modifying the metabolism criterion of clinical obesity (Supplementary Table 6). The proportion of adults with confirmed obesity classified as having clinical obesity varied strongly between different criteria. If any of the metabolic components were considered to indicate clinical obesity, the proportions of adults with confirmed obesity being clinical were 96.3% in NHANES and 98.5% in EPIC-Potsdam. Similar to the main analysis, the proportions of individuals with confirmed obesity classified as clinical were usually higher with increasing BMI level.

**Clinical obesity and risk of incident T2D and CVD**
Given that several clinical obesity criteria reflect major cardiometabolic risk factors, we next evaluated associations of preclinical and clinical obesity with risk of T2D and CVD in the EPIC-Potsdam study.

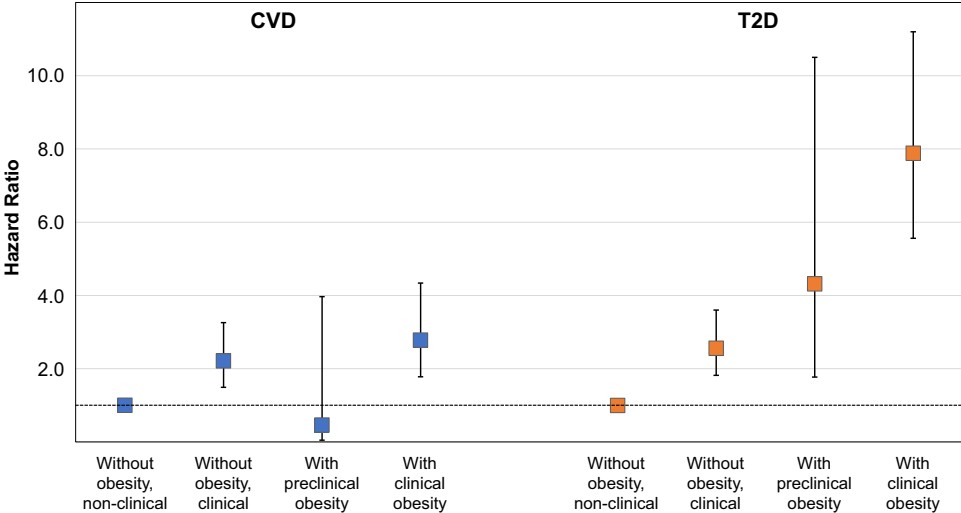

**Fig. 3 | Association of preclinical and clinical obesity with CVD and T2D incidence in the EPIC-Potsdam cohort.** Point estimates represent hazard ratios. Error bars represent 95% confidence intervals. Associations are adjusted for age, sex, sport and cycling, smoking status, educational status, alcohol consumption, consumption of whole grain, red meat, and coffee. Group sizes for CVD analyses: without obesity, non-clinical (not fulfilling clinical criteria) (reference group) $n = 560$, without obesity, clinical (fulfilling clinical criteria) $n = 1604$, with preclinical obesity $n = 29$, with clinical obesity $n = 413$; for T2D analyses: without obesity, non-clinical (reference group) $n = 558$, without obesity, clinical $n = 1692$, with preclinical obesity $n = 29$, with clinical obesity $n = 652$. CVD cardiovascular diseases, T2D type 2 diabetes.

Participants with clinical obesity had a substantially higher risk for T2D compared to participants without obesity not fulfilling clinical criteria (HR, 7.88, 95% CI: 5.56–11.2). Still, the risk was also high among participants with preclinical obesity (4.32, 1.77–10.5), and even individuals without obesity but who fulfilled clinical criteria had elevated risk (Fig. 3). With regard to CVD, preclinical obesity was not related to risk. In contrast, a more than doubled risk was observed among participants with clinical obesity (HR, 2.78, 95% CI: 1.78–4.34). Noteworthy, the risk among participants without obesity who fulfilled clinical criteria was substantially higher as well (2.21, 1.49–3.26).

The strict definition of the metabolic component (hyperglycemia in combination with low HDL-cholesterol and high triglyceride levels) likely assigns a preclinical rather than clinical state to many individuals who already show body fatness-related metabolic dysfunctions. We therefore repeated these analyses, using a reference group of individuals without obesity who were free of any metabolic signs (individuals with any metabolic alteration were included in the non-obese clinical group). HRs increased for CVD for all groups which met clinical criteria compared to the initial analysis (Supplementary Table 7).

### Response to lifestyle interventions

We then investigated in TULIP the effects of weight loss on the change of the proportion of clinical obesity, the changes of the individual components (hypertension, metabolic, and renal) that were used for the diagnosis of clinical obesity, and selected parameters that may predict remission of clinical obesity. In 150 people with a BMI ≥ 30 kg/m² the median weight loss during 9 months of the lifestyle intervention was 3.0% and people where then divided into a group with ≤ 3% weight loss ($n = 75$) and a group with > 3% weight loss ($n = 75$). Participants of the group that lost ≤ 3% of body weight had a median weight change of -0.99% during 9 months of follow-up (Supplementary Table 8). At baseline, 65% of these participants had clinical obesity and most of them were classified with clinical obesity based on the cardiovascular (hypertension) criterion. A total of 13% fulfilled the metabolic criterion. Only few anthropometrics and clinical parameters improved during the intervention. Remission of clinical obesity was observed in 9 out of 49 participants, and as a result, the proportion of clinical obesity decreased from 65% to 61%. A decrease of the cardiovascular and

metabolic, but not the renal components of clinical obesity, that were used for the definition of clinical obesity, were observed. Among selected parameters, only age was associated with remission of clinical obesity, with borderline significance (Supplementary Table 9).

Participants of the group that lost > 3% of body weight had a median weight change of -7.4% (Table 2). At baseline, 71% of the participants had clinical obesity, with most meeting the cardiovascular (hypertension) criterion. A total of 13% fulfilled the metabolic criterion. Most anthropometrics and clinical parameters improved during the intervention. Remission of clinical obesity was observed in 14 out of 53 participants and, as a result, the proportion of clinical obesity decreased from 71% to 57%. A decrease of all three components of clinical obesity that were used for the definition of clinical obesity (cardiovascular, metabolic, and renal) was observed. Noteworthy, among the components of the metabolism criterion, substantial reductions in triglyceride levels (from 148 to 118 mg/dl, -20%) and the proportion with prediabetes (from 52% to 29%) were observed, while HDL-cholesterol levels did not change. Among selected parameters, higher age, higher liver fat content and higher visceral fat mass were associated with a lower chance of remission of clinical obesity, while gender, BMI at baseline or change in weight were not significantly associated. In addition to age, liver fat content, but not visceral fat mass, explained large part of the variability in remission of clinical obesity (Supplementary Table 10).

Sensitivity analyses grouping participants according to a 5% weight loss-cutoff ($n ≤ 5%$ weight loss $= 104$, $n > 5%$ weight loss $= 46$) showed mostly comparable results (Supplementary Tables 11–14). However, the small size of the > 5% weight-loss subgroup limits the conclusions that can be drawn.

### Discussion

For almost all adults with BMI-defined obesity, obesity was confirmed by at least one additional anthropometric measure. While our findings from NHANES 2017-2018 confirm those recently published from this survey[5], we additionally validated this observation in a European population-based cohort. Thus, our study strongly supports the notion that confirmation of a BMI-based obesity status by other anthropometric measures as suggested by the *Commission*[2] is not informative. Clearer guidance on which anthropometric measures and cut-offs to

**Table 2 | Anthropometric and metabolic characteristics and fulfillment of criteria of clinical obesity before and after a 9-months lifestyle intervention in adults with BMI ≥ 30 kg/m² and > 3% decrease in body weight, Tübingen Lifestyle Intervention Programme**

| Parameter | Baseline | Follow-up | p-value |
|---|---|---|---|
| Gender (males/females) | 30/45 | 30/45 | NA |
| Age (years) | 49 (25–69) | 50 (25–70) | <0.0001 |
| Body weight (kg) | 100.9 (66.5–166.0) | 93.5 (62.7–160.7) | <0.0001 |
| BMI (kg/m²) | 34.8 (30.1–45.7) | 32.2 (26.1–44.1) | <0.0001 |
| Waist circumference (cm) | 109.2 (85–150) | 102.6 (83–139) | <0.0001 |
| Systolic blood pressure (mmHg) | 130.7 (96–181) | 124.7 (95–153) | 0.0003 |
| Diastolic blood pressure (mmHg) | 79.3 (49–108) | 75.4 (52–99) | 0.002 |
| eGFR (ml/min/1.73 m²) | 87.0 (51–160) | 89.2 (51–194) | 0.11 |
| Triglycerides (mg/dl) | 148.4 (41–857) | 117.8 (49–383) | <0.0001 |
| HDL-cholesterol (mg/dl) | 50.6 (33–82) | 49.8 (33–80) | 0.82 |
| Visceral fat $_{1H-MRT}$ (kg) [a] | 4.2 (1.1–10.1) | 3.2 (0.8–7.7) | <0.0001 |
| Liver fat $_{MRS}$ (%) [b] | 9.2 (0.8–30.1) | 5.5 (0.1–50.0) | 0.0003 |
| Prediabetes, n (%) | 39 (52) | 22 (29) | 0.0001 |
| Blood pressure-lowering medication, n (%) | 30 (40) | 27 (36) | <0.0001 |
| Lipid-lowering medication (%) | 5 (7) | 8 (11) | <0.0001 |
| Criteria of clinical obesity fulfilled | | | |
| Cardiovascular, n (%) | 51 (68) | 42 (56) | <0.0001 |
| Metabolic, n (%) | 10 (13) | 5 (7) | 0.0003 |
| Renal, n (%) | 3 (4) | 2 (3) | 0.04 |
| Clinical obesity, n (%) | 53 (71) | 43 (57) | <0.0001 |

Data represent unadjusted medians (range). Differences between baseline and follow-up were tested using the matched pairs *t*-test for continuous data and Fisher's exact test for categorical data. One-sided tests were used for pairwise comparisons of continuous parameters (for increase in age, HDL-cholesterol, and eGFR, and for decrease of all other continuous parameters), and two-sided tests were used for categorical parameters.
*MRT* magnetic resonance tomography, *MRS* magnetic resonance spectroscopy, *eGFR* estimated glomerular filtration rate.
[a]Available in 59 subjects at baseline and 51 at follow-up.
[b]At baseline available in 54 and at follow-up in 46 subjects.

use is needed to identify clinically important subgroups within the obesity population.

This work provides estimates for the prevalence of clinical obesity among US adults based on a representative survey, building upon previous reports that relied on BMI-based diagnostic criteria[5,9–11]. The estimated proportion of adults with confirmed obesity who fulfilled any diagnostic criteria increased with increasing BMI and age. This suggests that differentiating between clinical and subclinical obesity might be most informative among younger adults and those with class I obesity. However, our data from the European population-based EPIC-Potsdam study support that a high proportion of clinical obesity would need to be expected also in populations with a substantially lower relative proportion of participants with more severe forms of obesity compared to NHANES (in NHANES, about 50% of individuals with obesity had a BMI ≥ 35.0 kg/m², while this proportion was only ~20% in EPIC-Potsdam). Overall, the majority of individuals being classified by BMI to have obesity would also be classified as having clinical obesity.

Our results also indicate that most individuals with clinical obesity meet multiple defined clinical criteria. Restricting the diagnostic criteria to components that identify distinct groups rather than overlapping risk profiles could substantially reduce the time and cost burden of diagnostic workup. Notably, ~70% of adults with obesity met the cardiovascular criterion alone in NHANES. The Commission[2] suggests that the diagnosis of clinical obesity should only be made after excluding other causes of organ/tissue dysfunction. This approach adds complexity because metabolic and cardiovascular criteria, for example, are not specific to obesity. In clinical practice, it may be difficult to rule out that hyperglycemia, dyslipidemia, and hypertension are related to factors other than obesity, such as diet, activity levels, alcohol, and tobacco use. Conducting such differential diagnoses is impractical for most diagnostic criteria in health surveys such as NHANES and in observational population-based studies such as EPIC-Potsdam. Thus, assessing the prevalence of clinical versus preclinical obesity at the population level presents significant challenges.

Clinical obesity was prospectively related to a substantially higher risk of T2D and CVD. However, the risk was also substantially raised among individuals without obesity who fulfilled clinical criteria. In contrast to T2D, where obesity increases risk per se, the preclinical obesity group had a similar CVD risk to individuals without obesity but meeting clinical criteria. These findings largely mirror those observed for other classifications of obesity subgroups, specifically metabolic healthy obesity[12]. However, risk in BMI subgroups does depend on risk factors and cut-offs used for the definition of subgroups[12,13]. Our results from evaluating alternative criteria suggest here that the metabolic criteria, as defined by the *Commission*, classify individuals with obesity-related metabolic dysfunction as preclinical despite their increased cardiometabolic risk. The *Commission* defined the metabolism criterion of clinical obesity as the simultaneous presence of hyperglycemia, high triglyceride levels, and low HDL-cholesterol levels[2]. This approach differs from other definitions used to identify clinically important subgroups among adults with obesity. For instance, metabolic and bariatric surgery guidelines do not require specific comorbidities for individuals with BMI ≥ 35 kg/m² and recommend surgery in patients with T2D and a BMI ≥ 30 kg/m²[14]. Metabolic unhealthy obesity is often defined by the presence of multiple but not all components of the metabolic syndrome[12]. We have previously reported that in NHANES-III, 90% of individuals with BMI-defined obesity had at least one component of the metabolic syndrome[6]. Furthermore, hyperglycemia, high blood pressure, or a high waist-to-hip ratio were sufficient to distinguish high-risk from low-risk individuals with obesity, with the low-risk group accounting for 41% of all individuals with obesity in NHANES-III[6]. We observed now that using alternative definitions of the metabolic criteria (either hyperglycemia or dyslipidemia) resulted in large shifts in the proportion classified as having clinical obesity. Important in this context are also the results of TULIP, showing that moderate weight loss substantially reduces high triglyceride levels and the proportion of adults with prediabetes. These data together indicate that many more adults with obesity-related metabolic dysfunctions would benefit from weight loss interventions in terms of their metabolic health and their long-term risk than the relatively few who fulfill the metabolism criterion according to the *Commission*.

We observed that weight loss > 3% brought about by a structured lifestyle intervention with modification of the diet and an increase in physical activity decreased the proportion of clinical obesity from 71% to 57%, with remission of clinical obesity observed in 14 out of the 53 participants. Lower age and liver fat content, but not BMI at baseline nor the amount of weight loss, independently predicted remission of clinical obesity in people with weight loss > 3%. Future studies including larger patient numbers, a larger range of BMI (in our TULIP subgroup, about ¾ had BMI < 37 kg/m²) and achieving more pronounced weight loss by lifestyle intervention or other treatment options will be helpful to disentangle if further staging of clinical

obesity, considering for example existing BMI categories, is informative to account for potentially different treatment response patterns. Investigating ectopic fat accumulation in organs other than the liver (e.g., pancreatic fat) would also be of interest, given its relationship with cardiometabolic risk beyond BMI[15].

Our study has several limitations. First, while we used various measures of body fat to confirm a BMI-based diagnosis of obesity according to the *Commission's* criteria[2], some cut-offs are not well established in this context. Specifically, the use of a waist-to-height ratio of 0.5 to indicate obesity is debatable, as already pointed out by the *Commission*. This cut-off might better reflect increased risk associated with overweight (about 95% of adults with a BMI 25.0-29.9 kg/m² had a waist-to-height ratio > 0.5 in NHANES) rather than strictly indicating obesity. The National Health Service in England suggests that a waist-to-height ratio of 0.5 indicates "increased central adiposity", while a ratio of 0.6 or above corresponds to "high central adiposity"[16], which might align more closely with the health risks associated with a BMI ≥ 30 kg/m². However, using 0.6 instead of 0.5 had minimal impact on the proportion of individuals with obesity confirmed by any additional criteria in NHANES. Secondly, we used a random sub-cohort in EPIC-Potsdam to investigate the relative proportions of preclinical and clinical obesity and a relatively small subpopulation of the TULIP intervention study to investigate remission of clinical obesity. This might have limited precision given the sample sizes. Thirdly, our results provide evidence regarding the risk association of the newly proposed obesity definitions with incident CVD, defined as MI or stroke. While heart failure is a diagnostic component of clinical obesity, it would still be informative to investigate risk of heart failure across differently defined preclinical/clinical obesity groups given its strong association with obesity[17]. Such analyses were not possible in EPIC-Potsdam due to unavailability of relevant baseline clinical information for incident heart failure cases. Fourthly, information on impaired glucose tolerance, as a relevant component of dysglycaemia that can identify at-risk individuals not captured by HbA1c or fasting glucose, was unavailable in NHANES and EPIC-Potsdam. Fifthly, in addition to the modest sample size, the TULIP intervention study had no true control group. The intra-individual variability of fasting plasma glucose and 2-hour OGTT glucose limits the ability to assess remission of clinical obesity as being the consequence of weight loss in TULIP. Larger weight-loss intervention trials including a control group would be required to minimize misclassification of remission and reduce the risk of misinterpretation. Lastly, misclassification of clinical obesity status was possible due to the use of self-reported diagnoses and symptoms, the inability to determine if signs and symptoms were consequences of obesity (as large proportions of adults without obesity also fulfilled criteria), and the unavailability of indications for organ dysfunctions, e.g., related to the central nervous and lymphatic system. This limitation allowed classification of clinical obesity based on nine in NHANES 2017-2018 and only four in EPIC-Potsdam, out of the twelve clinical components defined by the *Commission*[2]. While recent studies on cancer risk in the UK Biobank covered a larger set of clinical components compared to EPIC-Potsdam, classification of preclinical and clinical obesity was largely based on diagnosis codes from hospital inpatient records[7,8]. Consequently, individuals with clinical obesity but without a hospital admission were misclassified as subclinical obesity in these studies. Generally, not distinguishing between dysfunctions caused by obesity and those due to other causes would tend to overestimate the proportion of clinical obesity. Conversely, not applying the full set of diagnostic criteria would tend to underestimate this proportion. The lack of complete coverage of diagnostic criteria in studies and the resulting misclassification is likely a major obstacle for comparative research to support their implementation into practice.

In conclusion, based on NHANES data from US adults and EPIC-Potsdam data from German adults, almost all adults with BMI-based obesity had at least one other anthropometric indicator of adiposity that were recommended by the *Commission* and at least four out of five adults with confirmed obesity met the newly proposed criteria for clinical obesity. The prevalence of clinical obesity increased with age and BMI. The proposed metabolic criterion may classify some individuals as having preclinical rather than clinical obesity, despite their high-risk cardiometabolic phenotype, and modifying the metabolic criteria resulted in a wide variability in clinical prevalence. Weight loss by lifestyle intervention led to remission of clinical obesity independent of baseline BMI, but affected individual components of the metabolism criterion substantially stronger. Therefore, refinement of clinical criteria is important to account for the likely significant variation in risk and treatment benefit within this group.

## Method
### NHANES 2017-2018
NHANES is a series of nationally representative surveys of US adults. Beginning in 1999, NHANES has been a continuous, multistage, nationally representative survey of the noninstitutionalized civilian resident US population. Data collected through in-home interviews and study visits at mobile examination centers have been released in 2-year cycles. This study included the 2017-2018 cycle due to the availability of measures reflecting obesity (weight, height, waist and hip circumference, DXA-measured body fat percentage) and diagnostic criteria of clinical obesity. The unweighted overall response rate was 51.9% for the interview component and 48.8% for the examination component. 5265 participants aged 20 years or older were included. The National Center for Health Statistics Research Ethics Review Board approved NHANES. Written informed consent was obtained from all adult participants.

Information on age, gender, race and ethnicity, medication use, and medical conditions was collected during household interviews according to fixed-category questions. Race and ethnicity were included because of known racial and ethnic differences in obesity prevalence. Weight, height, waist circumference, hip circumference, body fat, blood pressure, and liver stiffness were measured in mobile examination centers with standard protocols by trained personnel. The mean of the second and third available blood pressure measurements was used to calculate systolic and diastolic blood pressure. The percentage of body fat was measured using the Hologic Discovery model A densitometers (Hologic, Inc., Bedford, Massachusetts), software version Apex 3.2. and analyzed with Hologic APEX version 4.0 software. Liver stiffness was measured via FibroScan® model 502 V2 Touch. HbA1c was measured via HPLC. HDL-cholesterol levels were directly measured. Triglyceride and glucose levels were measured through enzymatic methods in a random subset of the participants who fasted for 8 to less than 24 h. Urine albumin and creatinine levels were measured with a fluorescent immunoassay and the Jaffe rate reaction method, respectively. Serum creatinine levels were measured using the Jaffe kinetic rate method. Estimated glomerular filtration rate (eGFR) was computed according to the Chronic Kidney Disease Epidemiology Collaboration equation[18]. FIB-4 and the NAFLD fibrosis score were calculated based on age, alanine aminotransferase, and aspartate aminotransferase (both measured via kinetic rate reaction), platelet count (measured using the Beckman Coulter DxH 800 instrument), and, for the NAFLD fibrosis score, additionally albumin and BMI[19,20]. Detailed descriptions of the assessment methods are publicly accessible[21].

### EPIC-Potsdam study
The EPIC-Potsdam study is a prospective cohort that recruited from 1994 to 1998 27,548 participants (16,644 women and 10,904 men; primarily of Middle European ancestry, age range: 35–65 years) from the general population of Potsdam and surroundings, Germany[22]. Follow-up occurred every 2–3 years via mailed questionnaires, with

response rates ranging from 90% to 96%. The study protocol was approved by the ethics committee of the Medical Society of the State of Brandenburg, Germany, and all participants provided a statement of written informed consent before enrollment.

Incident CVD was defined as the incidence of primary nonfatal and fatal MI and stroke (International Statistical Classification of Diseases and Related Health Problems (ICD)-10 codes: I21 for acute MI, I63.0 to I63.9 for ischemic stroke, I61.0 to I61.9 for intracerebral hemorrhage, I60.0 to I60.9 for subarachnoid hemorrhage and I64.0 to I64.9 for unspecified stroke). Incidence of CVD was captured by participants' self-reports or based on information from death certificates. Information on the incidence of T2D (ICD-10 code E11) was systematically acquired through self-report of a diagnosis, T2D-relevant medication, or dietary treatment due to a T2D diagnosis during follow-up. Additionally, death certificates and information from tumor centers, physicians or clinics that provided assessments for other diagnoses were screened for indication of incident T2D. For participants who were classified as potential cases based on that information, standard inquiry forms were sent to the treating physician to provide diagnostic details on CVD and T2D. Only physician-verified cases with a diagnosis date after the baseline examination were considered confirmed incident cases of CVD and T2D.

Nested case–cohorts were constructed for efficient study of molecular phenotypes. From all participants who provided blood at baseline ($n = 26{,}437$), a random sample (subcohort, $n = 2500$) was drawn, which served as the population for estimating proportions with confirmed obesity and clinical obesity ($n = 2391$ after excluding individuals where the clinical obesity state could not be determined due to missing information) and as a common reference population for both T2D and CVD endpoints. For each endpoint, all incident cases that occurred in the full cohort until a specified censoring date (30 November 2006 for CVD, 31 August 2005 for T2D) were included in the analysis. We excluded participants with prevalent or non-verifiable outcomes ($n_{CVD} = 124$, $n_{T2D} = 59$), possible CVD cases according to WHO MONICA criteria ($n_{CVD} = 38$), self-reported diabetes or diabetic medication at baseline ($n_{T2D} = 60$), missing follow-up information ($n_{CVD} = 56$, $n_{T2D} = 60$), missing BMI ($n_{CVD} = 5$, $n_{T2D} = 5$) or where fulfillment of clinical criterion could not be determined ($n_{CVD} = 119$, $n_{T2D} = 131$). The final CVD case-cohort sample comprised 2606 participants, including 403 cases (48 internal). The final T2D case-cohort sample comprised 2904 participants, including 785 cases (73 internal). Follow-up was defined as the time between enrollment and study exit, determined by diagnosis of the respective disease, death, dropout, or final censoring date, whichever came first.

Information on administrative sex and date of birth of the participants was forwarded by the residence registry. Anthropometric and blood pressure measurements were conducted according to standardized protocols[23,24]. Information on lifestyle and education was obtained using questionnaires and computer-assisted personal interviews. These included information on recreational physical activity, smoking status, average alcohol intake, diet, and educational attainment. Blood was partitioned into serum, plasma (with 10% of total volume citrate) and blood cells and was subsequently separately stored in tanks of liquid nitrogen at -196 °C or in deep freezers at -80 °C until the time of analysis. HbA1c and plasma concentrations of HDL-cholesterol, triglycerides, and glucose were measured at the Department of Internal Medicine, University of Tübingen, with an automatic ADVIA 1650 analyzer (Siemens Medical Solutions) in 2007. All biomarker measurements conducted in plasma were corrected for the dilution introduced by citrate volume to improve comparability with concentrations measured in EDTA-plasma reported in the literature.

## TULIP study

Individuals at high risk for T2D from the southern part of Germany participated in TULIP, a prospective uncontrolled diet and exercise intervention study (supported by the German Research Foundation (DFG) and registered on their website - https://gepris.dfg.de/gepris/projekt/5396893?language=en). The study was performed at the University Hospital of Tübingen, Germany. The study started in 2003 and the lifestyle intervention of the last patient included in the study ended in 2009. Outcomes of the study – e.g. glucose tolerance, insulin sensitivity, insulin secretion, body fat distribution, liver fat content, adipokines, hepatokines, genetic risk of T2D etc. – have been published[25–27]. People were included in the study when they fulfilled at least one of the following criteria: a family history of T2D ; BMI > 27 kg/m²; impaired glucose tolerance (2-h blood glucose ≥ 7.78 mmol/l); or previous diagnosis of gestational diabetes. They were considered healthy according to a physical examination and routine laboratory tests. The participants had no history of liver disease and did not consume more than 2 alcoholic drinks per day. Exclusion criteria were the diagnosis of diabetes mellitus or the presence of a severe, critical mental or physical illness. Informed written consent was obtained from all participants and the Ethics Committee of the University of Tübingen, Germany, approved the protocol.

A total of 413 individuals were enrolled. For this analysis, we included TULIP participants who had a BMI ≥ 30 kg/m² at baseline, with available information on diagnostic criteria for clinical obesity, who underwent 9 months of intensive lifestyle intervention ($n = 150$). At baseline, gender was assessed via self-report, medical history was obtained, and a physical examination was performed. Phenotypic characterization was performed using a 75 g OGTT, whole-body MRI and H-MRS, as well as a cycle exercise test. This examination was repeated after 9 months. After the baseline examination, all participants underwent monthly sessions of standardized individual face-to-face dietary counselling by trained diabetes educators. Counselling aimed to reduce body weight by ≥ 5%. Furthermore, participants were advised to reduce fat intake to < 30% of total energy consumed and to increase dietary fiber intake to at least 15 g/4185 kJ (1000 kcal). They were also encouraged to reduce the intake of saturated fatty acids to < 10% of total fat. During each visit, participants presented a 3-day food diary and discussed it with the dietitians. Diet composition was estimated with validated software. Individuals were motivated to perform at least 3 h of moderate exercise per week. A heart rate monitor was given to the participants, and they were instructed to exercise so that their heart rate was just below their individual anaerobic threshold. Individual anaerobic threshold was measured by a motorized treadmill.

## Definition and confirmation of obesity

BMI was calculated by dividing weight in kilograms by height in meters squared. The waist-to-hip ratio was calculated as the ratio of waist and hip circumference (both in cm) and the waist-to-height ratio as the ratio of waist circumference and height (both in cm). Similar to Aryee et al.[5], we used ethnic-specific cut-offs for BMI and waist circumference according to the *Commission* report[2] and World Health Organization[28–30]. Obesity was defined as a BMI ≥ 30 kg/m² except for non-Hispanic Asian, for whom obesity was defined as BMI ≥ 27.5 kg/m². Accordingly, obesity class I was defined as BMI 30.0–34.9 kg/m² (27.5–32.4 kg/m² in non-Hispanic Asian), obesity II as BMI 35.0–39.9 kg/m² (32.5–< 37.5 kg/m²), and obesity class III as BMI ≥ 40.0 kg/m² ( ≥ 37.5 kg/m²). Following the *Commission* report on clinical obesity[2], we confirmed high body fatness among those meeting the BMI threshold of obesity using additional anthropometric measurements: waist circumference (for non-Hispanic Asians ≥ 80 cm for women and ≥ 90 cm for men, otherwise ≥ 88 cm for women and ≥ 102 cm for men), waist-to-hip ratio ( > 0.85 for women, > 0.90 for men), waist-to-height ratio ( > 0.5 for both sexes and genders), and DXA-based body fat percentage ( > 35% for women, > 25% for men)[31,32]. Participants with BMI-based obesity who met any of the above criteria were considered to have "confirmed obesity".

## Definition of preclinical and clinical obesity

Factors to categorize individuals with confirmed obesity into either preclinical or clinical obesity were based on those proposed by the *Commission*[2]. A detailed list of indications of obesity-related organ or tissue dysfunction proposed by the *Commission* and the corresponding information available in NHANES 2017-2018 and EPIC-Potsdam is provided in Supplementary Table 15. Briefly, in NHANES we included signs and symptoms related to 1) upper airways: adapted STOP-Bang score[33], 2) respiratory system: self-reported shortness of breath, 3) cardiovascular system: self-reported heart failure, elevated measured blood pressure, self-reported antihypertensive medication, or self-reported shortness of breath 4) metabolism: hyperglycemia (self-reported diabetes, impaired fasting glucose or high HbA1c) combined with low HDL-cholesterol levels and high triglyceride levels or lipid-lowering medication 5) renal: microalbuminuria (urinary albumin-creatinine ratio $\geq$ 30 mg/mmol) or reduced eGFR ($< 60$ mL/min per 1.73 m$^2$)[34] or received dialysis in past 12 months, 6) urinary system: self-reported chronic urinary incontinence, 7) liver: hepatic fibrosis (FIB-4 index[35,36], increased NAFLD Fibrosis Score[19,35], or increased hepatic stiffness measured by vibration-controlled transient elastography (FibroScan®)), 8) musculoskeletal: self-reported osteoarthritis, and 9) self-reported limitations of activities of daily living (ADL) if reported to be due to weight problems. Indications of obesity-related dysfunction specific to the central nervous, reproductive, and lymphatic systems were unavailable. In EPIC-Potsdam, we included 1) cardiovascular system: self-reported heart failure, elevated measured blood pressure ( >130/80 mmHg), or self-reported antihypertensive medication 2) metabolism: hyperglycemia (self-reported diabetes, impaired fasting glucose or high HbA1c) combined with low HDL-cholesterol levels and high triglyceride levels or lipid-lowering medication, 3) renal: reduced eGFR[34], and 4) reproductive: tried to become pregnant for more than one year in females, sought medical consultation on fertility in males. In the TULIP study, we considered the following clinical criteria: 1) cardiovascular system (blood pressure > 130/80 mmHg or anti-hypertensive medication), 2) metabolism (impaired fasting glucose, impaired glucose tolerance or high HbA1c [prediabetes] combined with low HDL-cholesterol levels [or self-reported lipid-lowering medication] and high triglyceride levels [or self-reported lipid-lowering medication]), and 3) renal (reduced eGFR [ < 60 ml/min/1.73 m$^2$][34]).

## Statistical analysis

The NHANES 2017-2018 cycle was used to estimate prevalences of confirmed obesity and of preclinical and clinical obesity, overall and by age (20–44, 45–64, and $\geq$ 65 years), gender (men and women), and race and ethnicity (non-Hispanic White, non-Hispanic Black, Hispanic overall, Mexican American as a separate Hispanic subgroup, non-Hispanic Asian, and other). Stratified analyses according to previously described subgroups and the presence of clinical obesity (yes/no) were conducted. We also estimated proportions among adults across different BMI categories who fulfilled single or multiple diagnostic criteria for clinical obesity. These analyses accounted for the complex sampling design through inclusion of survey weights and design variables according to NHANES analytic guidelines[21] to obtain weighted estimates representative of the total civilian noninstitutionalized US population. Missingness in the medical examination sample in variables relevant for analyses ranged from 0.9% in body weight to 34.6% in DXA-measured total body fat (Supplementary Table 16). As recommended by the NHANES analytic guideline[21], multiple imputation (m = 5) was applied using random forest to approximate missing data in the medical examination sample. Survey-weighted estimates were summarized using Rubin's rules considering intra- and inter-imputation variability.

In EPIC-Potsdam, we estimated proportions among adults who fulfilled criteria for confirmed obesity, using BMI and additional anthropometric measures, and who fulfilled single or multiple diagnostic criteria for clinical obesity using the random subcohort. We assessed the association between preclinical and clinical obesity and incident CVD and T2D with Cox proportional hazards regression within the case-cohorts, accounting for the design by assigning weights as proposed by Prentice and handling ties by Efron approximation. Age was the underlying time scale, with entry time as age at baseline and exit time as age at event or censoring. The fully adjusted model included age (as underlying time scale) sex, sport and cycling (hours/week), smoking status (never, former, current light < 20 units/day, current high $\geq$ 20 units/day), educational status (current in training/no certificate/part, skilled worker, professional school, college or higher education/university), alcohol consumption (never, former, current $\leq$ 12 g (women)/$\leq$ 24 g (men), current > 12 g (women)/ > 24 g (men)), consumption of whole grain, red meat, and coffee (all in g/day) as covariates.

In TULIP, differences between baseline and follow-up data were tested using the matched pairs *t*-test for continuous data and the Fisher's exact test for categorical data. Remission of clinical obesity, thus moving from clinical obesity (confirmed obesity with at least one clinical criterion met) at baseline to a state without meeting any clinical criterion at follow-up, was first investigated in univariate nominal logistic regression models. The parameters of interest were gender, age, BMI, visceral fat mass, and liver fat content at baseline and change in body weight, which we found to be important determinants of metabolic health in obesity[12]. Remission of clinical obesity was then investigated in multivariable nominal logistic regression models that included parameters with a statistically significant univariable association with remission. A stepwise forward logistic regression model was used to determine the variability in remission of clinical obesity explained by its determinants.

Data were analyzed with SAS (version 9.4), R Studio (version 2023.3.0.386), and JMP 16.2.0 (SAS Institute Inc, Cary, NC, USA).

## Reporting summary

Further information on research design is available in the Nature Portfolio Reporting Summary linked to this article.

## Data availability

Data supporting the National Health and Nutrition Examination Survey (NHANES) 2017-2018 analyzes are publicly available. Data from the EPIC (European Prospective Investigation Into Cancer and Nutrition)-Potsdam study and the Tuebingen Lifestyle Intervention Programme (TULIP) study are not publicly available due to data protection regulations. Data analyzes of EPIC-Potsdam and TULIP data in accordance with German Federal and State data protection regulations may be timeline unlimited initiated upon an informal enquiry addressed to the secretariat of the DIfE Human Study Center (Office.HSZ@dife.de) and the TULIP Data Use and Access Committee (projektantrag.idm@med.uni-tuebingen.de). Each request will then have to pass a formal process of application and review by the respective Principal Investigator and a scientific board. Source data are provided with this paper.

## Code availability

The scripts for data analyses have been uploaded to Zenodo (https://doi.org/10.5281/zenodo.17860390).

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

## Acknowledgements

This study was supported by a grant from the German Federal Ministry of Research, Technology and Space (BMFTR) to the German Center for Diabetes Research (DZD) (institutional funding; 82DZD03D03, C.S. and M.B.S.) and the State of Brandenburg and furthermore supported by funding from the European Innovative Health Initiative SOPHIA (875534, N.S.). The funders had no role in the design and conduct of the study.

## Author contributions

C.S. had full access to NHANES and EPIC-Potsdam data and takes responsibility for the integrity and accuracy of the data analysis. N.S. had full access to the TULIP data and takes responsibility for the integrity of the data and the accuracy of the data analysis. Concept and design: C.S., F.H., M.B.S., and N.S.; Interpretation of data: C.S., F.H., M.B.S., and N.S.; Drafting of the manuscript: C.S., M.B.S., and N.S.; Critical revision of the manuscript for important intellectual content: C.S., F.H., M.B.S., and N.S.; Statistical analysis: C.S. and N.S.

## Funding

## Competing interests

The authors declare no competing interests.
