## [Transparent Peer Review file · Nature Communications]

Prevalence of Clinical Obesity, Association with Cardiometabolic Risk and Response to Lifestyle Intervention

Corresponding Author: Professor Matthias Schulze

Version 0:

Reviewer comments:

Reviewer #1

(Remarks to the Author)

The authors responses to reviewer comments were adequate. Specifically, we want to highlight the appropriateness and potential areas for improvement for some of the responses as part of this review.

1. We agree with the reviewer's concern that the paper appeared fragmented without a clear rationale for why those specific datasets (NHANES, EPIC, and TULIP) and not others were used was adequately addressed by the authors' response. They now explicitly link each dataset (NHANES for prevalence, EPIC-Potsdam for prospective risk, TULIP for intervention response) to the corresponding study aims and provide supporting references to prior related work. The revised manuscript reflects this structure clearly, and I find the response satisfactory.
2. The reviewer wanted a stronger rationale for Aims 2 and 3, noting that the metabolic healthy/unhealthy obesity literature already addresses a similar concept. The authors have acknowledged the overlap with the metabolic healthy/unhealthy obesity literature but appropriately clarified that their scope is to evaluate the newly proposed Lancet DE Commission's definition of preclinical/clinical obesity. While comparison with other classifications is of interest, it is indeed beyond the scope of this paper. The revised Introduction now more clearly justifies Aims 2 and 3 in terms of assessing long-term cardiometabolic risk and treatment response. I find the response satisfactory.
3. The reviewer raised a concern about novelty since part of the analysis had already been published in a JAMA paper. The authors acknowledge that part of their analysis (confirmation of obesity in NHANES) replicates findings from Aryee et al. in JAMA. However, they clearly distinguish the novel contribution of their study in applying the second and third aims. They have also moved the overlapping material to the supplement to keep the main focus on new results. I find this a reasonable and satisfactory response.
4. These changes address the reviewer's concerns and improve methodological consistency. I find the responses satisfactory.
5. The authors have clarified the definitions and corrected a small error in the renal criteria, specified that limitations in daily activities were only included when explicitly linked to weight problems (noting the very small impact of this component), and acknowledged the limitations in available measures within NHANES and EPIC. They appropriately frame the latter as both a limitation and a reflection of real-world challenges in implementing the new classification. I find the responses satisfactory.
6. The reviewer's critique seems to conflate the purpose of this study with the novelty of the concept itself. The "clinical obesity" classification was proposed by the Lancet DE Commission, not by these authors, and the goal of this paper is to evaluate that framework empirically. The authors acknowledge overlap with prior metabolic healthy/unhealthy obesity research but clarify that their aim was to evaluate the new Lancet DE Commission classification of clinical obesity, which uses distinct criteria. In this context, demonstrating the risk associations of the new classification is an important and novel contribution, even if the general direction of findings is consistent with prior literature. I find the response satisfactory.

7. The reviewer's suggestion is reasonable — obesity is strongly related to heart failure (HF), often more than stroke/MI, so including HF as an outcome could strengthen the analysis. The authors argue that heart failure cannot be evaluated prospectively because it is already part of the clinical obesity criteria, which I agree would create overlap if prevalent HF were included at baseline. However, incident HF occurring after baseline could still provide valuable information, given the strong relationship between obesity and HF. If such data are unavailable, this limitation could be acknowledged more explicitly.

8. The reviewer flagged the case-cohort design itself as a limitation. The authors appropriately clarify that the case-cohort design is a well-established and efficient method for prospective analyses, and not a limitation per se. They now acknowledge that the limited sample size of the sub-cohort may affect the precision of prevalence estimates in EPIC-Potsdam, which is a reasonable framing. This is a solid response.

9. The reviewer pointed out that there are larger and more established CVD cohorts that might be better suited for the analyses done by the authors. The authors acknowledge that those cohorts exist and are valuable. They then argued that those cohorts do not necessarily capture all the clinical criteria needed for the Lancet DE Commission's definition of clinical obesity. Their stance is reasonable as the purpose here isn't just to study CVD in general — it's to evaluate the new definition of clinical obesity. That requires data on both anthropometric and metabolic dysfunction criteria, which are not systematically collected in all cohorts.

10. The reviewer is concerned that using T2D as an outcome might seem circular, since diabetes (or hyperglycemia) is part of the criteria for clinical obesity. The authors included self-reported T2D as part of the hyperglycemia criterion when estimating the prevalence of clinical obesity but excluded prevalent T2D cases from prospective analyses to focus only on incident, physician-confirmed diagnoses. This approach aligns with the Lancet DE Commission's framing of hyperglycemia as both a diagnostic component and a risk state.

11. The reviewer expressed concerns about the small sample size, lack of control, and regression to the mean, which might have explained the findings. The authors acknowledge regression to the mean and have added analyses using participants with minimal weight loss as a comparator. However, this does not fully resolve the concern that remission of prediabetes may largely reflect variability in fasting or 2-hr glucose measures, which are known to fluctuate. While the deep phenotyping in TULIP is a strength, the limitations of sample size, lack of a true control group, and reliance on single glucose measurements for remission should be more clearly acknowledged.

12. The reviewer pointed out the existence of larger, randomized weight loss trials that might be more suitable for studying remission than TULIP. The authors highlight the detailed phenotyping in TULIP and argue that many large contemporary weight loss trials often lack a true placebo group. However, this does not completely address the concern that bigger randomized trials with structured comparator groups and broader clinical criteria could be better suited to study remission within the clinical obesity framework. While TULIP offers valuable mechanistic insights, its small sample size (only 150 out of 413 recruited individuals were included) and absence of a control group remain significant limitations. Specifically, the lack of a comparator arm restricts the ability to distinguish genuine remission effects from regression to the mean, especially given the variability in fasting and 2-hour glucose measures as previously noted. We recommend the authors acknowledge this limitation in the manuscript.

13. The authors justify the use of a 3% threshold on the basis that it reflected the mean weight loss in TULIP, with the >3% group achieving a mean loss of 7.4%. However, this rationale feels somewhat arbitrary, and it would be more compelling to align with the conventional 5% benchmark for clinically meaningful weight loss. At minimum, presenting both 3% and 5% cut-offs would address the reviewer's concern and situate the findings more clearly in the clinical context.

14. The authors have revised their NHANES analyses to apply race- and ethnicity-specific cut-offs for BMI and anthropometric measures, as recommended by the Commission. They report only minor impact on prevalence estimates and appropriately justify their use of standard cut-offs in the European cohorts (EPIC-Potsdam, TULIP). I find this response adequate.

15. Additional Comments:

i) In the Introduction, the authors state that "...Both preclinical and clinical obesity groups may harbor subgroups of substantially different cardiometabolic risk and different requirements and goals of intervention, but evidence necessary for further staging and assignment of treatment is unavailable yet." This phrasing could be misinterpreted to suggest that the present study directly compares treatment responses between preclinical and clinical obesity. Since the TULIP analyses focus on individuals with obesity (BMI \geq 30) and remission of clinical obesity, without a clear subgroup comparison of preclinical vs clinical obesity, I suggest clarifying this point to avoid reader confusion.

ii) Please ensure that all NHANES cycles are labeled as "2017–2018." There is an error in the results section that shows NHANES 2017-2028.

Reviewer #2

(Remarks to the Author)

Reviewer #3

(Remarks to the Author)

1. ORIGINALITY AND SIGNIFICANCE:

What I understand as the core of this manuscript is assessing the cardiovascular and metabolic import of the classifications of clinical/subclinical obesity described by a Lancet Diabetes and Endocrinology Commission.

- Firstly the inclusion of several populations (NHANES , EPIC Potsdam) is unnecessarily confusing and not clearly justified.
- The inclusion of an intervention study does not address the main aim of the study. It really should have been the subject of another manuscript.

A manuscript is not a grant, so it can not have multiple aims unless they are parallel and closely aligned. The interventional aspect of this manuscript does not contribute to the message of the first part of the manuscript.

2. DATA AND METHODOLOGY:

Since the Lancet Diabetes and Endocrinology commission definition of clinical obesity and pre clinical obesity are being utilized, these must be explicitly outlined. This is not done in the methods, nor is it done in the methods section. The sentences added to the introduction are much too vague.

3. CONCLUSIONS :

While this point is reasonable, there is no conclusion possible since the aims of the manuscript are so disparate and non aligned.

4. SUGGESTED IMPROVEMENTS:

This manuscript needs to be more cohesive, and this is only possible if the focus is either on the metabolic/cardiovascular import of the Lancet Diabetes and Endocrinology commission definitions, or the effect of a weight loss intervention on the Lancet Diabetes and Endocrinology commission categories.

5. CLARITY AND CONTEXT:

The revision retains the same issue: no central theme . You simply have to chose ONE aim and express it clearly , stick with one population and have a clear take away message. I am not convinced including all aims in one manuscript resulted in “important contributions to the discussion around this novel obesity classification concept”, rather it makes your manuscript difficult to understand, and having no unified message.

Reviewer #4

(Remarks to the Author)

1. Although the results show significant implications for clinical practice, the authors did not present suggestions on how to better improve the current classification based on the robust data here presented; Suggestion - A more detailed exploration of the sensitivity of the results to different definitions of clinical obesity would strengthen the analysis;

RESPONSE: We have included sensitivity analyses for the metabolism criterion of the clinical obesity definition. We used five alternative definitions to compare the proportion of individuals fulfilling the clinical obesity criteria and found relevant differences in the proportional assignment (eTable 6). We have also previously investigated the impact of modifying the criteria (metabolism component) on risk associations (eTable 7). However, our main focus is to evaluate the classification as close as possible to the laid-out criteria – not to provide an alternative classification. We hope the Reviewer agrees that a systematic comparison of alternative definitions across all components included in the definition and their quantification with cardiometabolic risk would go far beyond the scope of this work.

Response from Reviewer: I agree that the authors have presented alternative definitions/sensitivity analysis for the metabolism criteria in eTable 6, which demonstrated important differences between the criteria. I would request that the authors more clearly define definition #5 in the Table - what is meant by the term ‘Original definition’? Requires edit or Table footnotes.

One further question – why is impaired glucose tolerance not used as a criterion in eTable 6? Since this is an important component of dysglycaemia, reflecting different aetiology and hence defining a different cohort vs. that defined by IFG.

I would also recommend a minor edit in line 6 of the Abstract, with addition of the word ‘metabolic’: “More than 80% of adults with confirmed obesity met the criteria for clinical obesity, a proportion influenced by age, BMI and metabolic criteria used”.

I agree with the authors comment that, whilst of interest, a systematic comparison of different definitions of clinical obesity would likely expand this manuscript beyond its current scope.

2. The cross-sectional nature of the data does not allow to draw strong conclusions on the associated cardiometabolic risk.

RESPONSE: Analyses in NHANES are cross-sectional, the analyses in EPIC-Potsdam and TULIP are, however, prospective. The analyses in EPIC-Potsdam encompass a median follow-time of ~8 years for CVD and ~6 years for T2D and the follow-up procedures are described in the Methods section. We assume that this was a misunderstanding.

Response from Reviewer: Agree. To make this clear I would suggest include brief details re study design in the Abstract. This would add only 4 or 5 additional words. If there is possibility of further expanding the Abstract I would suggest that the cohort sizes are also included.

The number of participants from the TULIP study is very limited, hence data using larger longitudinal cohorts would be more suitable for the results and impact of the data here presented. Can the authors add data from these longitudinal cohorts with a greater N?

RESPONSE: We used prospective data from the longitudinal EPIC-Potsdam cohort study, which includes 27,548 participants. The embedded case-cohorts were used to estimate cardiometabolic risk associations as described in detail in the Methods section. If referring to TULIP as intervention study, we agree that data from larger weight loss trials could be analyzed. However, we have focused on a study where precise baseline phenotyping was performed in an obese population that underwent a structured lifestyle intervention with repeated measurements of these phenotypes. This allowed us to carefully characterize the participants regarding body fat mass and distribution, liver fat content, insulin sensitivity and insulin secretion, parameters that are closely related to the definition of clinical obesity. To increase the sample size, we now performed the analyses in the whole TULIP study population, including participants who lost $\leq 3\%$ of body weight. We agree with the reviewer that, thereby, we could provide a broader picture about the changes of parameters defining clinical obesity during a structured lifestyle intervention.

Response from Reviewer: I note the additional analyses run by the authors which is a good addition, however the sample size for both cohorts in TULIP is still small. The subgroup for wt loss $>3\%$, $n=75$ and wt loss $<3\%$, $n=75$. I find that confusing since earlier publications of TULIP presented much larger sample size of $n=357$ at 9 month follow up. doi 10.1007/s00125-017-4407-z. Please explain how the small $n=150$ sub-cohort was selected.

I believe that this model could be more rigorously tested using a larger cohort, preferably also with a Control group, for which there are many trials published with well characterized baseline criteria similar to TULIP. Please include this small sample size of the TULIP study into the limitations section on pages 12/13, and the need to evaluate a larger more robust cohort in future analyses.

3. Also, the TULIP study did not have a control group; that must be addressed adequately (I would suggest replacing this cohort with a more robust study - greater N and control group);

RESPONSE: We did not include a control arm in the TULIP study because we specifically focused on determinants of a successful lifestyle intervention in people at risk of type 2 diabetes. Thus, each participant received the same amount of counselling and motivation to lose weight, which allowed us to study the individual metabolic response to the intervention and the parameters determining this response. In practice, contemporary obesity trials would also unlikely include one arm being pure "no treatment" or "pure placebo" with zero lifestyle or behavioral support. TULIP also did not require specific high-risk states to be present, e.g. prediabetes. Thus, participants reflect a broader spectrum of individuals with obesity. We have used TULIP as data source quite extensively before to investigate treatment response patterns in the context of metabolic phenotypes (e.g. PMIDs: 19889804, 20841609, 21174075, 20042761, 27185609, 28919065 28840257). Still, we now include data also from those participants with smaller or no weight loss (see eTables 8 and 9).

Response from Reviewer: Thank you for that addition. I would still propose a larger cohort with Control group would be preferable.

4. There was significant heterogeneity on the variables used to classify clinical obesity, which could lead to misinterpretation of the results. For instance, I would not use self-reported data in this analysis or, at least, the authors could do some sensitivity analysis to address this.

RESPONSE: The definition of clinical obesity by the Lancet DE Commission includes self-reported (e.g. self-reported medical conditions, medication or impairment of daily activities) as well as directly measured information (e.g. blood pressure). We have been following recommendations of the Commission as close as possible. That research studies have not been designed so far to systematically assess all clinical criteria is a point we discuss - different data availability unavoidably leads to heterogeneity how classifications are conceptualized in studies. We have been very transparent about our approaches in this context (e.g. NHANES variable list in eTable 11) and don't think that sensitivity analyses would be specifically helpful. What we have done in the revision is a close alignment of the criteria in our different studies (e.g. alignment of blood pressure criteria).

Response from Reviewer: No further comments

5. Besides liver fat, pancreas fat might actually be an important predictor of cardiometabolic risk. There are cohorts with measures of pancreatic fat that could be useful to complete this analysis (). This is just a note for thought. I understand that practically, liver fat is easier to assess from a clinical perspective.

RESPONSE: We agree with the reviewer that pancreatic fat content is also a predictor of cardiometabolic risk, yet less strong than liver fat content (we performed cross-sectional analyses addressing pancreatic fat in Tübingen, but don't have many data in people who had follow-up analyses of pancreatic fat content).

Response from Reviewer: I would recommend including cohorts with MR-assessment of pancreatic fat in future analyses. Ectopic fat is likely important.

6. Suggestion: Explicitly state which potential confounding variables were considered and adjusted for in the analyses (in the methods section).

I have also added minor notes to the manuscript, that can be found in the attached document.

RESPONSE: We have added information on the adjustment variables for the TULIP study in the Methods section. The adjustment variables for the prospective analyses in EPIC-Potsdam are already listed there. Prevalence analyses in NHANES did not include adjustments as confounding bias is irrelevant in this context. Since the attached document was not forwarded, the additional notes cannot be addressed at this stage.

Response from Reviewer: No further comments

Reviewer #5

(Remarks to the Author)

Version 1:

Reviewer comments:

Reviewer #1

(Remarks to the Author)

No further comments

Reviewer #4

(Remarks to the Author)

Thank you to the authors for the revisions made to the manuscript, and the point by point responses to my questions/suggestions. I believe that it would be informative if the authors choose to look at larger intervention studies in the future, but no further changes to this manuscript suggested.

Reviewer #5

(Remarks to the Author)

REVIEWER COMMENTS

Reviewer #1 (Remarks to the Author):

The authors responses to reviewer comments were adequate. Specifically, we want to highlight the appropriateness and potential areas for improvement for some of the responses as part of this review.

1. We agree with the reviewer's concern that the paper appeared fragmented without a clear rationale for why those specific datasets (NHANES, EPIC, and TULIP) and not others were used was adequately addressed by the authors' response. They now explicitly link each dataset (NHANES for prevalence, EPIC-Potsdam for prospective risk, TULIP for intervention response) to the corresponding study aims and provide supporting references to prior related work. The revised manuscript reflects this structure clearly, and I find the response satisfactory.

Revision Response: Thank you, no changes were made.

2. The reviewer wanted a stronger rationale for Aims 2 and 3, noting that the metabolic healthy/unhealthy obesity literature already addresses a similar concept. The authors have acknowledged the overlap with the metabolic healthy/unhealthy obesity literature but appropriately clarified that their scope is to evaluate the newly proposed Lancet DE Commission's definition of preclinical/clinical obesity. While comparison with other classifications is of interest, it is indeed beyond the scope of this paper. The revised Introduction now more clearly justifies Aims 2 and 3 in terms of assessing long-term cardiometabolic risk and treatment response. I find the response satisfactory.

Revision Response: Thank you, no changes were made.

3. The reviewer raised a concern about novelty since part of the analysis had already been published in a JAMA paper. The authors acknowledge that part of their analysis (confirmation of obesity in NHANES) replicates findings from Aryee et al. in JAMA. However, they clearly distinguish the novel contribution of their study in applying the second and third aims. They have also moved the overlapping material to the supplement to keep the main focus on new results. I find this a reasonable and satisfactory response.

Revision Response: Thank you, no changes were made.

4. These changes address the reviewer's concerns and improve methodological consistency. I find the responses satisfactory.

Revision Response: Thank you, no changes were made.

5. The authors have clarified the definitions and corrected a small error in the renal criteria, specified that limitations in daily activities were only included when explicitly linked to weight problems (noting the very small impact of this component), and acknowledged the limitations in available measures within NHANES and EPIC. They appropriately frame the latter as both a limitation and a reflection of real-world challenges in implementing the new classification. I find the responses satisfactory.

Revision Response: Thank you, no changes were made.

6. The reviewer's critique seems to conflate the purpose of this study with the novelty of the concept itself. The "clinical obesity" classification was proposed by the Lancet DE Commission, not by these authors, and the goal of this paper is to evaluate that framework empirically. The authors acknowledge overlap with prior metabolic healthy/unhealthy obesity research but clarify that their aim was to evaluate the new Lancet DE Commission classification of clinical obesity, which uses distinct criteria. In this context, demonstrating the risk associations of the new classification is an important and novel contribution, even if the general direction of findings is consistent with prior literature. I find the response satisfactory.

Revision Response: Thank you, no changes were made.

7. The reviewer's suggestion is reasonable — obesity is strongly related to heart failure (HF), often more than stroke/MI, so including HF as an outcome could strengthen the analysis. The authors argue that heart failure cannot be evaluated prospectively because it is already part of the clinical obesity criteria, which I agree would create overlap if prevalent HF were included at baseline. However, incident HF occurring after baseline could still provide valuable information, given the strong relationship between obesity and HF. If such data are unavailable, this limitation could be acknowledged more explicitly.

Revision Response: We agree that investigating incident HF in preclinical obesity (thus, transition to clinical obesity based on a single criterion) and also among those with clinical obesity but without heart failure at baseline would be interesting additional research questions. However, while we do have detailed biomarker profiles measured in the CVD and T2D case-cohorts, this is unfortunately not the case for incident HF cases. As a consequence, relevant information for the clinical obesity classification, such as HbA1c and blood glucose measurements, are not available for incident HF cases in the EPIC-Potsdam cohort. Omitting these parameters of the criteria would induce misclassification and reduce comparability with the other analyses. We therefore did not perform additional analyses but highlighted this point in the limitations section (p13-14).

8. The reviewer flagged the case-cohort design itself as a limitation. The authors appropriately clarify that the case-cohort design is a well-established and efficient method for prospective analyses, and not a limitation per se. They now acknowledge that the limited sample size of the sub-cohort may affect the precision of prevalence estimates in EPIC-Potsdam, which is a reasonable framing. This is a solid response.

Revision Response: Thank you, no changes were made.

9. The reviewer pointed out that there are larger and more established CVD cohorts that might be better suited for the analyses done by the authors. The authors acknowledge that those cohorts exist and are valuable. They then argued that those cohorts do not necessarily capture all the clinical criteria needed for the Lancet DE Commission's definition of clinical obesity. Their stance is reasonable as the purpose here isn't just to study CVD in general — it's to evaluate the new definition of clinical obesity. That requires data on both anthropometric and metabolic dysfunction criteria, which are not systematically collected in all cohorts.

Revision Response: Thank you for supporting this point, we didn't make any additional changes.

10. The reviewer is concerned that using T2D as an outcome might seem circular, since diabetes (or hyperglycemia) is part of the criteria for clinical obesity. The authors included self-reported T2D as

part of the hyperglycemia criterion when estimating the prevalence of clinical obesity but excluded prevalent T2D cases from prospective analyses to focus only on incident, physician-confirmed diagnoses. This approach aligns with the Lancet DE Commission's framing of hyperglycemia as both as diagnostic component and a risk state.

Revision Response: Thank you, no changes were made.

11. The reviewer expressed concerns about the small sample size, lack of control, and regression to the mean, which might have explained the findings. The authors acknowledge regression to the mean and have added analyses using participants with minimal weight loss as a comparator. However, this does not fully resolve the concern that remission of prediabetes may largely reflect variability in fasting or 2-hr glucose measures, which are known to fluctuate. While the deep phenotyping in TULIP is a strength, the limitations of sample size, lack of a true control group, and reliance on single glucose measurements for remission should be more clearly acknowledged.

Revision Response: We agree and clearly state those points now in the limitation section (p13-14).

12. The reviewer pointed out the existence of larger, randomized weight loss trials that might be more suitable for studying remission than TULIP. The authors highlight the detailed phenotyping in TULIP and argue that many large contemporary weight loss trials often lack a true placebo group. However, this does not completely address the concern that bigger randomized trials with structured comparator groups and broader clinical criteria could be better suited to study remission within the clinical obesity framework. While TULIP offers valuable mechanistic insights, its small sample size (only 150 out of 413 recruited individuals were included) and absence of a control group remain significant limitations. Specifically, the lack of a comparator arm restricts the ability to distinguish genuine remission effects from regression to the mean, especially given the variability in fasting and 2-hour glucose measures as previously noted. We recommend the authors acknowledge this limitation in the manuscript.

Revision Response: We agree and clearly state those points in the limitation section now (p13-14).

13. The authors justify the use of a 3% threshold on the basis that it reflected the mean weight loss in TULIP, with the >3% group achieving a mean loss of 7.4%. However, this rationale feels somewhat arbitrary, and it would be more compelling to align with the conventional 5% benchmark for clinically meaningful weight loss. At minimum, presenting both 3% and 5% cut-offs would address the reviewer's concern and situate the findings more clearly in the clinical context.

Revision Response: We now performed sensitivity analyses using 5% weight-loss cut-offs for comparison and added them to the Supplement (Supplementary Tables 11-14) with a short summary in the results section. The results show comparable results as when using 3% weight loss as the cut-off. However, the small size of the >5% weight-loss subgroup (n= 46) limits the conclusions that can be drawn.

14. The authors have revised their NHANES analyses to apply race- and ethnicity-specific cut-offs for BMI and anthropometric measures, as recommended by the Commission. They report only minor impact on prevalence estimates and appropriately justify their use of standard cut-offs in the European cohorts (EPIC-Potsdam, TULIP). I find this response adequate.

Revision Response: Thank you, no changes were made.

15. Additional Comments:

i) In the Introduction, the authors state that "...Both preclinical and clinical obesity groups may harbor subgroups of substantially different cardiometabolic risk and different requirements and goals of intervention, but evidence necessary for further staging and assignment of treatment is unavailable yet." This phrasing could be misinterpreted to suggest that the present study directly compares treatment responses between preclinical and clinical obesity. Since the TULIP analyses focus on individuals with obesity (BMI ≥ 30) and remission of clinical obesity, without a clear subgroup comparison of preclinical vs clinical obesity, I suggest clarifying this point to avoid reader confusion.

Revision Response: We slightly adapted this sentence exclusively referring to clinical obesity. It now reads: Clinical obesity groups may harbor subgroups of substantially different cardiometabolic risk and different requirements and goals of intervention, but evidence necessary for further staging and assignment of treatment is unavailable yet.

ii) Please ensure that all NHANES cycles are labeled as "2017–2018." There is an error in the results section that shows NHANES 2017-2028.

Revision Response: Thank you, we corrected the typo.

Reviewer #3 (Remarks to the Author):

1. ORIGINALITY AND SIGNIFICANCE:

What I understand as the core of this manuscript is assessing the cardiovascular and metabolic import of the classifications of clinical/subclinical obesity described by a Lancet Diabetes and Endocrinology Commission.

- Firstly the inclusion of several populations (NHANES , EPIC Potsdam) is unnecessarily confusing and not clearly justified.
- The inclusion of an intervention study does not address the main aim of the study. It really should have been the subject of another manuscript.

A manuscript is not a grant, so it can not have multiple aims unless they are parallel and closely aligned. The interventional aspect of this manuscript does not contribute to the message of the first part of the manuscript.

2. DATA AND METHODOLOGY:

Since the Lancet Diabetes and Endocrinology commission definition of clinical obesity and pre clinical obesity are being utilized, these must be explicitly outlined. This is not done in the methods, nor is it done in the methods section. The sentences added to the introduction are much too vague.

3. CONCLUSIONS :

While this point is reasonable, there is no conclusion possible since the aims of the manuscript are so disparate and non aligned.

4. SUGGESTED IMPROVEMENTS:

This manuscript needs to be more cohesive, and this is only possible if the focus is either on the metabolic/cardiovascular import of the Lancet Diabetes and Endocrinology commission definitions, or the effect of a weight loss intervention on the Lancet Diabetes and Endocrinology commission categories.

5. CLARITY AND CONTEXT:

The revision retains the same issue: no central theme . You simply have to chose ONE aim and express it clearly , stick with one population and have a clear take away message. I am not convinced including all aims in one manuscript resulted in “important contributions to the discussion around this novel obesity classification concept”, rather it makes your manuscript difficult to understand, and having no unified message.

Revision Response: We disagree with several aspects of this review. The main criticism repeatedly raised is the lack of focus. Integrating closely related aims within a single manuscript is standard practice and allows a coherent, comprehensive assessment of the research question, maintains consistent definitions, and analytic choices across datasets, and avoids splitting connected results across multiple papers. Combining complementary study designs enables methodological triangulation: NHANES cross-sectional analyses provide nationally representative U.S. prevalence estimates, but incidence information is largely unavailable. Prospective analyses in EPIC-Potsdam address this by estimating incident disease risks and establishing temporality. The TULIP intervention study evaluates responsiveness to preventive treatment and potential remission, thereby, despite its limitations, complementing the observational evidence. Taken together, combining those designs provide more meaningful insights into the potential impact of the newly proposed definition than any single design alone.

Reviewer #4 (Remarks to the Author):

1. Although the results show significant implications for clinical practice, the authors did not present suggestions on how to better improve the current classification based on the robust data here presented; Suggestion - A more detailed exploration of the sensitivity of the results to different definitions of clinical obesity would strengthen the analysis;

RESPONSE: We have included sensitivity analyses for the metabolism criterion of the clinical obesity definition. We used five alternative definitions to compare the proportion of individuals fulfilling the clinical obesity criteria and found relevant differences in the proportional assignment (eTable 6). We have also previously investigated the impact of modifying the criteria (metabolism component) on risk associations (eTable 7). However, our main focus is to evaluate the classification as close as possible to the laid-out criteria – not to provide an alternative classification. We hope the Reviewer agrees that a systematic comparison of alternative definitions across all components included in the definition and their quantification with cardiometabolic risk would go far beyond the scope of this work.

Response from Reviewer: I agree that the authors have presented alternative definitions/sensitivity analysis for the metabolism criteria in eTable 6, which demonstrated important differences between the criteria. I would request that the authors more clearly define definition #5 in the Table - what is meant by the term 'Original definition'? Requires edit or Table footnotes.

Revision Response: We exchanged 'Original definition' for 'Metabolism criterion defined as hyperglycemia (self-reported diabetes, impaired fasting glucose or high HbA1c), low HDL-cholesterol levels, and high triglyceride levels' to be more specific and improve clarity.

One further question – why is impaired glucose tolerance not used as a criterion in eTable 6? Since this is an important component of dysglycaemia, reflecting different aetiology and hence defining a different cohort vs. that defined by IFG.

Revision Response: Specific information on impaired glucose tolerance is unfortunately unavailable in NHANES 2017-2018 and EPIC-Potsdam. We support the point, that IGT is a relevant component of dysglycaemia that may identify at-risk individuals not identified by elevated HbA1c or fasting glucose measurements. We have therefore added this point to the limitations section (p. 13-14). Important to consider in this context is that OGTTs are in most countries not among the routine measurements conducted in clinical practice. Therefore, use of HbA1c and IFG as diagnostic measures may reasonably reflect the actual proportion of individuals that would be classified to be clinically obese in practice.

I would also recommend a minor edit in line 6 of the Abstract, with addition of the word 'metabolic': "More than 80% of adults with confirmed obesity met the criteria for clinical obesity, a proportion influenced by age, BMI and metabolic criteria used".

Revision Response: We agree with this point, but we have removed this section to comply with the journal's abstract word limit.

I agree with the authors comment that, whilst of interest, a systematic comparison of different definitions of clinical obesity would likely expand this manuscript beyond its current scope.

2. The cross-sectional nature of the data does not allow to draw strong conclusions on the associated cardiometabolic risk.

RESPONSE: Analyses in NHANES are cross-sectional, the analyses in EPIC-Potsdam and TULIP are,

however, prospective. The analyses in EPIC-Potsdam encompass a median follow-time of ~8 years for CVD and ~6 years for T2D and the follow-up procedures are described in the Methods section. We assume that this was a misunderstanding.

Response from Reviewer: Agree. To make this clear I would suggest include brief details re study design in the Abstract. This would add only 4 or 5 additional words. If there is possibility of further expanding the Abstract I would suggest that the cohort sizes are also included.

Revision Response: We now shortened the abstract according to the journal guidelines to 150 words but indicated the prospective nature of the analyses by wording, such as “incidence of cardiovascular diseases and type 2 diabetes”.

The number of participants from the TULIP study is very limited, hence data using larger longitudinal cohorts would be more suitable for the results and impact of the data here presented. Can the authors add data from these longitudinal cohorts with a greater N?

RESPONSE: We used prospective data from the longitudinal EPIC-Potsdam cohort study, which includes 27,548 participants. The embedded case-cohorts were used to estimate cardiometabolic risk associations as described in detail in the Methods section. If referring to TULIP as intervention study, we agree that data from larger weight loss trials could be analyzed. However, we have focused on a study where precise baseline phenotyping was performed in an obese population that underwent a structured lifestyle intervention with repeated measurements of these phenotypes. This allowed us to carefully characterize the participants regarding body fat mass and distribution, liver fat content, insulin sensitivity and insulin secretion, parameters that are closely related to the definition of clinical obesity. To increase the sample size, we now performed the analyses in the whole TULIP study population, including participants who lost $\leq 3\%$ of body weight. We agree with the reviewer that, thereby, we could provide a broader picture about the changes of parameters defining clinical obesity during a structured lifestyle intervention.

Response from Reviewer: I note the additional analyses run by the authors which is a good addition, however the sample size for both cohorts in TULIP is still small. The subgroup for wt loss $>3\%$, $n=75$ and wt loss $<3\%$, $n=75$. I find that confusing since earlier publications of TULIP presented much larger sample size of $n=357$ at 9 month follow up. doi 10.1007/s00125-017-4407-z. Please explain how the small $n=150$ sub-cohort was selected.

Revision Response: We are sorry that our description of TULIP in the manuscript confused the reviewer in light of previous publications. Please note that participants were included in the study when they fulfilled at least one of the following criteria: a family history of type 2 diabetes; BMI > 27 kg/m²; IGT (blood glucose ≥ 7.78 mmol/l); or previous diagnosis of gestational diabetes as we describe in the Methods section. Thus, obesity was not a requirement for inclusion. We restricted the analyses for this manuscript to those individuals with BMI ≥ 30 and who underwent the 9-month intervention, with 150 participants meeting these criteria. While this has been described in the Methods section before, we have now rearranged this section (p. 19) to make the flow of participants easier for readers to follow.

I believe that this model could be more rigorously tested using a larger cohort, preferably also with a Control group, for which there are many trials published with well characterized baseline criteria similar to TULIP. Please include this small sample size of the TULIP study into the limitations section on pages 12/13, and the need to evaluate a larger more robust cohort in future analyses.

Revision Response: We now clearly acknowledge the limitations of the TULIP study (p.13-14), including the small sample size, lack of a true control group, and the reliance on single glucose and 2h OGTT measurements for remission classification, in the limitations section of the discussion. We also state that a larger intervention study, addressing these issues, is needed to confirm the findings.

3. Also, the TULIP study did not have a control group; that must be addressed adequately (I would suggest replacing this cohort with a more robust study - greater N and control group);

RESPONSE: We did not include a control arm in the TULIP study because we specifically focused on determinants of a successful lifestyle intervention in people at risk of type 2 diabetes. Thus, each participant received the same amount of counselling and motivation to lose weight, which allowed us to study the individual metabolic response to the intervention and the parameters determining this response. In practice, contemporary obesity trials would also unlikely include one arm being pure “no treatment” or “pure placebo” with zero lifestyle or behavioral support. TULIP also did not require specific high-risk states to be present, e.g. prediabetes. Thus, participants reflect a broader spectrum of individuals with obesity. We have used TULIP as data source quite extensively before to investigate treatment response patterns in the context of metabolic phenotypes (e.g. PMIDs: 19889804, 20841609, 21174075, 20042761, 27185609, 28919065 28840257). Still, we now include data also from those participants with smaller or no weight loss (see eTables 8 and 9).

Response from Reviewer: Thank you for that addition. I would still propose a larger cohort with Control group would be preferable.

Revision Response: We accept that and included a statement on the limitations of the TULIP study in the discussion, also highlighting the need for larger intervention studies for confirmation.

4. There was significant heterogeneity on the variables used to classify clinical obesity, which could lead to misinterpretation of the results. For instance, I would not use self-reported data in this analysis or, at least, the authors could do some sensitivity analysis to address this.

RESPONSE: The definition of clinical obesity by the Lancet DE Commission includes self-reported (e.g. self-reported medical conditions, medication or impairment of daily activities) as well as directly measured information (e.g. blood pressure). We have been following recommendations of the Commission as close as possible. That research studies have not been designed so far to systematically assess all clinical criteria is a point we discuss - different data availability unavoidably leads to heterogeneity how classifications are conceptualized in studies. We have been very transparent about our approaches in this context (e.g. NHANES variable list in eTable 11) and don't think that sensitivity analyses would be specifically helpful. What we have done in the revision is a close alignment of the criteria in our different studies (e.g. alignment of blood pressure criteria).

Response from Reviewer: No further comments

Revision Response: Thank you, no changes were made.

5. Besides liver fat, pancreas fat might actually be an important predictor of cardiometabolic risk. There are cohorts with measures of pancreatic fat that could be useful to complete this analysis (). This is just a note for thought. I understand that practically, liver fat is easier to assess from a clinical perspective.

RESPONSE: We agree with the reviewer that pancreatic fat content is also a predictor of cardiometabolic risk, yet less strong than liver fat content (we performed cross-sectional analyses addressing pancreatic fat in Tübingen, but don't have many data in people who had follow-up analyses of pancreatic fat content).

Response from Reviewer: I would recommend including cohorts with MR-assessment of pancreatic fat in future analyses. Ectopic fat is likely important.

Revision Response: We now highlight the relevance of investigating pancreatic fat as a potential complementary marker in future studies in the discussion section (p 12).

6. Suggestion: Explicitly state which potential confounding variables were considered and adjusted for in the analyses (in the methods section).

I have also added minor notes to the manuscript, that can be found in the attached document.

RESPONSE: We have added information on the adjustment variables for the TULIP study in the Methods section. The adjustment variables for the prospective analyses in EPIC-Potsdam are already listed there. Prevalence analyses in NHANES did not include adjustments as confounding bias is irrelevant in this context. Since the attached document was not forwarded, the additional notes cannot be addressed at this stage.

Response from Reviewer: No further comments

Revision Response: Thank you, no changes were made.

Reviewer #5 (Remarks to the Author):

Revision Response: Thank you, we found your review very clear and constructive.